# Methods for Identifying Microbial Natural Product Compounds that Target Kinetoplastid RNA Structural Motifs by Homology and De Novo Modeled 18S rRNA

**DOI:** 10.3390/ijms22094493

**Published:** 2021-04-26

**Authors:** Harrison Ndung’u Mwangi, Edward Kirwa Muge, Peter Waiganjo Wagacha, Albert Ndakala, Francis Jackim Mulaa

**Affiliations:** 1Department of Biochemistry, University of Nairobi, Nairobi 00200-30197, Kenya; mugeek@uonbi.ac.ke (E.K.M.); mulaafj@uonbi.ac.ke (F.J.M.); 2Department of Computing and Informatics, University of Nairobi, Nairobi 00200-30197, Kenya; waiganjo@uonbi.ac.ke; 3Department of Chemistry, University of Nairobi, Nairobi 00200-30197, Kenya; andakala@uonbi.ac.ke

**Keywords:** kinetoplastids, 18S rRNA, homology and de novo modeling, natural products, virtual screening, molecular docking

## Abstract

The development of novel anti-infectives against Kinetoplastids pathogens targeting proteins is a big problem occasioned by the antigenic variation in these parasites. This is also a global concern due to the zoonosis of these parasites, as they infect both humans and animals. Therefore, we need not only to create novel antibiotics, but also to speed up the development pipeline for these antibiotics. This may be achieved by using novel drug targets for Kinetoplastids drug discovery. In this study, we focused our attention on motifs of rRNA molecules that have been created using homology modeling. The RNA is the most ambiguous biopolymer in the kinetoplatid, which carries many different functions. For instance, tRNAs, rRNAs, and mRNAs are essential for gene expression both in the pro-and eukaryotes. However, all these types of RNAs have sequences with unique 3D structures that are specific for kinetoplastids only and can be used to shut down essential biochemical processes in kinetoplastids only. All these features make RNA very potent targets for antibacterial drug development. Here, we combine in silico methods combined with both computational biology and structure prediction tools to address our hypothesis. In this study, we outline a systematic approach for identifying kinetoplastid rRNA-ligand interactions and, more specifically, techniques that can be used to identify small molecules that target particular RNA. The high-resolution optimized model structures of these kineoplastids were generated using RNA 123, where all the stereochemical conflicts were solved and energies minimized to attain the best biological qualities. The high-resolution optimized model’s structures of these kinetoplastids were generated using RNA 123 where all the stereochemical conflicts were solved and energies minimized to attain the best biological qualities. These models were further analyzed to give their docking assessment reliability. Docking strategies, virtual screening, and fishing approaches successfully recognized novel and myriad macromolecular targets for the myxobacterial natural products with high binding affinities to exploit the unmet therapeutic needs. We demonstrate a sensible exploitation of virtual screening strategies to 18S rRNA using natural products interfaced with classical maximization of their efficacy in phamacognosy strategies that are well established. Integration of these virtual screening strategies in natural products chemistry and biochemistry research will spur the development of potential interventions to these tropical neglected diseases.

## 1. Introduction

Kinetoplastids are a group of flagellated protozoans that are distinguished by the presence of a DNA-containing region, known as a “kinetoplast,” in their single large mitochondrion. These groups include a number of both animal and plant pathogens that are transmitted through different vectors and cause disease. Some of these pathogens that cause disease belong to the genus trypanosome and leishmania. *Trypanosoma cruzi* is the causative agent of chagas disease, *trypanosome brusei gambiense* and *trypanosome brusei rhodeense* are causative agents of Human African Trypanosomiasis (HAT), while *leishmania* causes leishmaniasis. From published data, nearly a billion individuals are at a risk of kinetoplastid pathogenic infection, with around 20 million reported cases worldwide, leading to over 95,000 deaths per year [1,2]. From statistics, leishmaniasis leads with an individual mortality rate of 50,000 per year and an annual loss of 2.1 million disability-adjusted life years (DALYs) [3]. This is followed by 48,000 deaths caused by HAT (which causes sleeping sickness) with 1.5 million more DALYs annually [1,4]. Last is Chagas disease, which causes 15,000 deaths and over 700,000 DALYs per year [1]. Despite the concerted efforts to combat these kinetoplastid infections, they continue to pose serious health and economic risks, particularly in endemic regions.

Existing therapeutics for diseases associated with these pathogens have limitations in toxicities, their associated costs, and their invasive administration routes; hence, they are not ideal. These could be shown for existing drugs for leishmaniasis treatment, amphotericin B deoxycholate and miltefosine [1,5]. HAT treatment drugs, eflornithine and pentamidine, in addition to the initial problems have varying efficacy in different disease stages and adverse severe side effects [1,6]. The use of nifurtimox and benznidazole for Chagas disease has been shown to be ideal for early stages of the disease but diminishes with the duration of the infection [7,8]. In addition to all these therapeutic factors for diseases caused by kinetoplastids, high attrition rate and resistance has been observed for many of the new emerging drugs with very poor penetration to remote areas; hence, there is an urgent need to develop novel intervenes with newer mechanisms of action.

## 2. RNA as a Drug Target

RNA is one of the most important macromolecule in the cell and a versatile chemical species in molecular biology [9]. RNA is involved in diverse roles in the cell, which include storage and transfer of genetic information, enzymatic catalysis, molecular recognition, and genetic regulation. For instance, tRNAs, rRNAs, and mRNAs are essential for gene expression both in the pro- and eukaryotes. However, all these types of RNAs have sequences and sometimes 3D structures that are specific to kinetoplastids only and can be used to shut down essential biochemical processes in the pathogen only. As a result, knowledge of the RNA three-dimensional structure has become a focal point for understanding its diverse biological functions [10].

One advantage of using kinetoplastid RNA as a drug target is that its secondary structure information, which includes the motifs that comprise an RNA, can be easily obtained from its sequence by free energy minimization [11] or phylogenic comparison. Furthermore, RNA motifs can have similar properties both as isolated systems and as parts of larger RNAs. Studies on the binding of aminoglycoside antibiotics to RNA loops have facilitated the development of compounds to combat multidrug resistance [12]. These results show that the identification of RNA motifs that bind small molecules are valuable for targeting the rRNAs that contain them [13]. However, since RNA can adopt diverse structures, including internal and hairpin loops, an understanding of how to target RNA using natural product molecules and other ligands has been elusive.

A complete understanding of these diverse biological functions of RNA molecules requires knowledge of their higher order structures, i.e., two-dimensional (2D) and three-dimensional (3D), as well as the characteristics of their primary sequence [14,15,16]. RNA structure is important for many of its functions, including the regulation of transcription and translation, catalysis, transport of proteins across membranes, and the regulation of RNA viruses. Understanding these functions is important for basic biology as well as for the development of drugs that can intervene in cases where pathological functionality of these molecules occurs [17]. Interactions are one of the most fundamental activities of biomolecules [10]. Disturbance of these interactions underlie biological disorders, including cancers and neurodegenerative diseases. Characterizing these interactions is important to understand the detailed mechanisms of life [9].

The purpose of molecular modeling is to provide functional insight in biological molecules, not to achieve some arbitrary precision in the atomic coordinates. Therefore, we seek to improve our abilities to construct 3D models for molecules for which we do not yet have experimental atomic-resolution structures and carefully identify the predicted features that yield important insights [10,18].

Advances in computer algorithms for ribosomal structure and function prediction have provided biologists with valuable information about their organelle of interest. Homology and de novo modeling has developed into a significant procedure in structural and functional biology that has served to narrow the gap between experimentally determined structures and known 18S rRNA sequences. Complete assembly and automation of algorithms have not only simplified the homology and de novo structure determination process, but also streamlined it to allow users to manually curate the modeling results, visualize, minimize energy, and interpret the result. This method of RNA structure determination coupled with de novo modeling has improved greatly and has contributed immensely to the functional insights of the 18S rRNA.

The process of discovering new promising substances that can be further developed to novel drugs involves undertaking structure determination coupled with virtual screening. These methods offer a more rational and direct approach to attaining low cost and high efficiency novel intervenes to these kinetoplastid diseases. Molecular docking and virtual screening methods are more prospective due to the ability to study the affinity between a particular natural compound bound to a specific 18S rRNA motif.

The mentioned aspects of 3D structure determination of the 18S rRNA coupled with molecular docking methods offer useful and promising methodologies of novel substance discovery that could be satisfactory remedies. In the current study, we combined the aforementioned strategies to predict high-resolution 3D structures of selected kinetoplastid 18S rRNA. This was further used to perform docking simulation with some natural compounds from myxobacteria. Our results reveal very informative molecular interaction that could be responsible for binding at specific 18S rRNA motifs.

## 3. Results and Discussion

### 3.1. Sequence selection Homology and De Novo Modeling Structure Prediction and Analysis of the 18S rRNA of the Selected Kinetoplastids Leishmania Major, Trypanosoma brucei, and Trypanosoma cruzi

We obtained the selected kinetoplastid genomes from the gene bank and aligned them with the whole database using BLASTn to search for the homologous genomes. After phylogenetic analysis and sequence alignment of all the genomes of the selected kinetoplastid species, the sequences were selected based on the completeness of each of the 18S sequenced genes deposited in the verified databases. The three best-ranked sequences, based not only on the completeness but also verified by other peer-reviewed researchers’ databases, were selected for alignments to assess the deviation in terms of genetic variation. This was performed for all the three kinetoplastids selected in this study, as shown in Table 1, Table 2 and Table 3 and Figure 1, Figure 2 and Figure 3 of the supplementary information. The consensus sequence indicated the similarity index of the sequences, but for completeness, this gave us the insight to pick the most complete index to be used as a guide through the whole process of structure prediction. It is important to note that the 18S rRNA structure is very conserved, with minimal allowed changes in the expansion segments, and since this is a single species, not much variation is expected. The selected kinetoplastid 18S rRNAs that had all nucleotides and were 100% complete were selected after rigorous analysis of the secondary structure prediction and eventual three-dimensional structure predictions. After all the analyses and alignments, using multi-align [19,20], were performed, the three selected genome sequences for the three kinetoplastids were obtained, as shown below in Table 1.

### 3.2. 18S rRNA Secondary Structure

The selected sequences of the three kinetoplastid species were taken through a rigorous exercise of determining their secondary structure using a software known as Varna [21]; (version 3.93 http://varna.lri.fr/) (accessed on 21 January 2021), RNAstructure [22]; (Version 6.1) https://rna.urmc.rochester.edu/RNAstructure.html) (accessed on 21 January 2021), RNAComposer [23] (http://rnacomposer.cs.put.poznan.pl/) (accessed on 21 January 2021), RNApdbee [24] (http://rnapdbee.cs.put.poznan.pl/) (accessed on 21 January 2021), xRNA **(**http://rna.ucsc.edu/rnacenter/xrna/) (accessed on 21 January 2021), and RNA2D3D [25] (https://binkley2.ncifcrf.gov/users/bshapiro/rna2d3d/rna2d3d.html) (accessed on 24 January 2021). Figure 4, Figure 5 and Figure 6 show the secondary structures of the 18S rRNA of *leishmania major* AC005806, *Trypanosoma brucei* M12676, and *Trypanosoma cruzi* AF245382 as predicted and verified in the Comparative RNA Web (CRW) Site [26].

### 3.3. Three-Dimensional Structures of the Modeled Kinetoplastids

We obtained the three-dimensional structures of the selected kinetoplastid, as shown in Figure 1, Figure 2 and Figure 3 below. Analysis and molecular modeling was performed to make sure the three structures conform to their biological function. This was done by further optimizing the structures to obtain further minimal energies, as shown in Table 2 below.

Just as with compound libraries, there are freely available libraries for 3D crystal structures of biological targets. These do not strictly consist of proteins, as there are DNA structures, nanoparticles, and peptides that have also been crystalized and are also available.

When searching for a target, the search box allows you to add a name of a compound; when you hit search, a large number of hits may be identified. To filter these hits and to make sure you have the correct target, the following criteria and filters should be met: the correct organism/taxonomy was selected followed by the correct strain (if organism); the experimental method should always be x-ray diffraction (however, NMR can be used if no other source is available); the resolution structure optimally should be below 2, which increases the accuracy of analyzing target; and the target should be deposited onto PDB recently (date should be most recent).

These criteria allowed us to have an updated and accurate 3D crystal structure of the target, which can now be used for molecular docking and/or molecular dynamic simulations. As with most chemical reactions, there are many factors that permit a stable system. In the protein data bank, there are crystalized targets with other small molecules at the active site and surrounded by solvent. The protein or nucleotides can then be subjected to various software, where all modifications can be made.

### 3.4. Myxobacterial Secondary Metabolites Databases

This is a growing source of secondary metabolites that were obtained from gram-negative proteobacteria [27]. These bacteria have a wide range of known habitats, which include decaying plant material, soil, tree barks, marine environments, and herbivore dung [27,28]. These naturally occurring microbes have several distinct characteristic behaviors, such as moving in solid surfaces by creeping and gliding, as amoebas do, which differentiate them from other bacteria [27,28,29]. In addition to this, they are known rich producers of natural secondary metabolites by virtue of their metabolism (i.e., *Bacillus species*, actinomycetes, Pseudomonads, and fungi) [30,31].

Close to 7500 strains of myxobacteria have provided at least 100 distinct core structures to date, but only a portion of these (67) have been reported in primary literature [27,32] alongside over 500 chemical derivatives [32,33]. Most Myxobacterial metabolites are non-ribosomal polypeptides, as well as their hybrids, polyketides, phenylpropanoids, alkaloids, and terpenoids [27,30,34,35]. Many strains of Myxobacterial metabolites belong to multiple structural classes in addition to the number of chemical variations on each scaffold [27,30]. Furthermore, many of the natural products reveal distinctive structural topographies comparative to compounds known from other microorganisms [30]. The best binding compounds to the kinetoplastids are shown in Table 4 of the supplementary information.

### 3.5. Binding and Docking Results

#### Selected Kinetoplastid 18S rRNA Structure Preparation

This study used the predicted structures of the 18S rRNA of the selected kinetoplastids *Leishmania major* (M12676)*, Trypanosoma brucei* (M12676)*,* and *Trypanosoma cruzi* (AF245382). The docked position of the best 20 compounds for all the kinetoplastids, as is shown below in Figure 4, Figure 5, Figure 6, Figure 7, Figure 8, Figure 9, Figure 10, Figure 11, Figure 12, Figure 13, Figure 14, Figure 15, Figure 16, Figure 17, Figure 18, Figure 19, Figure 20, Figure 21, Figure 22, Figure 23, Figure 24, Figure 25, Figure 26, Figure 27, Figure 28, Figure 29, Figure 30 and Figure 31. **A**: Best binding pose and nucleotides involved **B:** Shows schematic the binding pocket **C**: shows the Nucleotides component that re involved in binding **D**: Shows the main bonds involved between the compound and the nucleotide component

The molecular docking results gave several consensuses scoring value functions, which estimate the binding energies of study substances (Myxobacteria metabolites) with the 18S rRNA target obtained from both the Schrodinger and Accerlys discovery suite. The binding affinities in terms of the binding energies is shown as the atomic contact energies (ACE). The low values for the binding energies as per the software suites of the compounds docked to the 18S rRNA active motif sites gives a ligand pose in the actual active binding site. The binding site is an area where the hydrophobic fragment of the compound is engrossed, as shown in the images of the various poses for the same. The summarized table with the best poses for all three kinetoplastids is shown below. Of keen interest is previous work, which performed in vitro validation of *Mycobactrerium tuberculosis*, studying hybridization of the target–probe interaction (labelled MTB rRNA) on an antibiotic (Kenamycin and streptomycin) platform with a negative control (–ve Ctl; water) for the 18S rRNA. We suggest that this method could be further supplemented by the synthesis of aptamers for further analysis to qualify selected screening products as therapeutics.

### 3.6. Binding Site Identification

In some studies, the drug binding pocket on the biological target is unknown. In such situations, where it is impossible to dock compounds, it curtails further progress through the rational drug design process.

There are a huge number of online tools that are available to identify an active site from a protein; however, one of the most validated and popular tools is Metapocket (further reading in publications). Metapocket uses eight different algorithms to identify ligand binding sites by computing interactions between a chemical probe and a protein structure. The input is a PDB file of a protein structure, the output is a list of “interaction energy clusters” corresponding to putative binding sites. Table 4 shows the docking and binding site results of the best pose compounds with activity on all the kinetoplastids with more negative ACE −400 (48).

Computer-aided drug design can be broadly classified into two main subgroups:(1)Structure-based drug design: this method assumes that the structure of a biological target is known (the protein/DNA has been crystallized or a 3D model of the target is built). Compounds are then designed/screened to fit the structural characteristics of the target, thus rendering strong molecular interactions that stabilize the compound at the targets binding site. This technique is the most widely used in computational chemistry and yields a plethora of potential compounds that may then be screened for activity.(2)Ligand-based drug design: this method assumes that only the structure of the drug is known and that there is an absence of the 3D biological target. Optimized compounds are then designed based on the knowledge of the drug’s chemical analogs and their biological activity. Quantitative structure activity relationship (QSAR) features are designed based on physiochemical attributes of a set of chosen analogs and their biological activity with a target molecule. These QSAR features are then used as a template to screen for potential compounds with more favorable characteristics. Computational tools are now also available to predict potential targets of a compound prior to QSAR analysis.

Pharmacophore-based drug design, which implements aspects of both structure and ligand-based design, is an optimized and more accurate method of identifying optimized lead molecules. This method requires the 3D crystal structure of the target, as well as the structure activity relationship of the compound at the binding site of the target to be known. Once the intermolecular forces between the compound and target have been established, a pharmacophore model can be generated (pharmacophore—minimum number of atoms in a compound that is required to induce a biological response).

This pharmacophoric model/scaffold, containing only vital molecular moieties, is then used to screen chemical libraries to identify potential lead molecules.

### 3.7. Conclusion and Recommendation

In the present study, the 3D structure of the three selected kinetoplastid species were modeled and docked with the myxobacteria natural compounds from the database and the best 20 ligands are shown above (Figure 4, Figure 5, Figure 6, Figure 7, Figure 8, Figure 9, Figure 10, Figure 11, Figure 12, Figure 13, Figure 14, Figure 15, Figure 16, Figure 17, Figure 18, Figure 19, Figure 20, Figure 21, Figure 22, Figure 23, Figure 24, Figure 25, Figure 26, Figure 27, Figure 28, Figure 29, Figure 30 and Figure 31). The docking results identified 10 compounds as shown in Table 3, where motifs and areas of interaction with the ligands for the three 18S rRNA species of kinetoplastids with more negative atomic contact energies from ACE −400. This study has provided a methodology that has yielded a list of 10 compounds from myxobacteria that show activity against selected neglected tropical kinetoplastids. Further exploration of the activity of analogues of this compound is warranted to improve anti-parasitic selectivity together with in vitro screening using synthesized aptamers of the motifs. Target identification for the most promising compounds will support the future development of pan-active treatments against kinetoplastids.

Further compounds from this collection have potential as new chemical starting points for drug discovery efforts against one or more of the parasites tested.

## 4. Materials and Methods

### 4.1. Selection and Three-Dimentional Modeling of Kinetioplastids 18S rRNA

The sequences of the three kinetoplastids; *Leishmania major, Trypanosoma brucei*, and *Trypanosoma cruzi* 18S rRNA were obtained through blasting in the gene bank (NCBI) [36]. It is important to note that most of the sequences in the gene bank are not complete, so a further process of verification was required. We checked the completeness of the sequences using information available at specialized groups that do verification of 18S rRNA sequences. Such a group is The Comparative RNA Web (CRW) Site, which has a database that shows the completeness of sequences among other analyzed and verified annotation [26]. The sequences picked from this site were further analyzed to ascertain the sequences and minimize the errors. Back to the gene bank, the FASTA format of these refined sequences was picked and saved as text files using a notepad++ text editor **(**https://notepad-plus-plus.org/) (accessed on 15 December 2020). Since there is a possibility of having more than one complete sequence, the final query sequence to be modeled was obtained by carrying out further multiple sequence alignment to identify the one that deviated minimally from the consensus sequence. MultiAlign [19] software was used in the alignment to show how similar or dissimilar various sequences were.

### 4.2. Selecting a Template

Selection of template structures for the three kinetoplastid rRNA was a rigorous exercise that is described below. These required a search of various structure libraries using the query sequence.

### 4.3. Obtaining and Verifying Template Sequences and Three-Dimensional Coordinate Files

The templates for all the three kinetoplastids were again selected through an elaborate process that involved several steps. Firstly, by blasting individual query sequences in the gene bank (NCBI) and finding the sequences that are highly similar to the query and not in any way the query (we did not select the query if it was shown in the BLAST alignments) [37]. These sequences infer an evolutionary relationship with the query but not specifically the query if they cover a distinct region of the template to get a higher similarity score. Calculation of a local pair-wise alignment between the templates and their targets was performed. Secondly, checks and evaluations were done as the section above to check if these sequences were complete followed by a heuristic step, which intended to improve the alignment. Insertion and deletion placements in the template were considered for optimization. Of particular interest were the isolated residues in the alignment (Islands), which were moved to the flanks for the facilitation of loop building. The next step was to find out if there was any crystal structure of the 18S rRNA template sequences that were available. This was done by checking the Research Collaborators for Structural Bioinformatics (RCSB) Protein Data Bank (PDB) archive, which gives the 3D shapes of nucleic acids, proteins, and complex assemblies that help researchers and students understand all aspects of biology [38]. Coordinate files of the template structures of the 18S rRNAs obtained from the PDB website available online (http://www.rscb.org/pdb/explore.do) (accessed on 15 December 2020) were saved as PDB files on a text editor. Depending on the complexity of the rRNA homology, de novo modeling was done in parts by dividing it into the different parts after cleaning: 5′major, central, 3′minor, and 3′major domains. An important point noted was while splitting cut, the rRNA of both the template and the query at similar points to achieve the best structure at the end. The structure was viewed in different software available that can read PDB files, such as pymol and Accelrys, among others. The crystal structures of the templates obtained had some challenges such as unresolved portions and gaps and require to be further optimized.

### 4.4. Homology and De Novo Modeling

18S rRNA homology and de novo modeling were done using RNA123 version 2.0.1.3 and Genesilico software. RNA123 was able to predict the secondary and tertiary structure of the three kinetoplastids ribosomal RNA. RNA123 took three steps; Preprocessing, Alignment, and Modeling [18].

Curation and validation of the built model was performed using several validation tools, such as PRO-CHECK [39], MATCHCHECK [39], and MOL-PROBITY [40]. These software help to understand the stereochemistry and geometry of the 3D structure of the modeled 18S rRNA. Ramachandran plot statistics were used to evaluate the best model. The built model was further superimposed on the experimental crystal structure of the template and the root mean square deviation (RMSD) was calculated [41].

### 4.5. Ligand Docking Simulation

#### 4.5.1. Active Site Prediction

The active sites of the modeled 18S rRNA were predicted using discovery studio (Version 16.1.0.15350), which is based on the “Eraser” algorithm [42,43]. The active sites were further confirmed using MetaPocket 2.0 available at https://projects.biotec.tu-dresden.de/metapocket/index.php (accessed on 17 January 2021).

#### 4.5.2. Preparation of 18S rRNA and Ligand Molecules for Docking

The 18S rRNA model were prepared for docking in “Preparation Wizard” (Schrodinger suite version 2018-4) using default settings. The three-dimensional SDF files of myxobacteria natural compounds used as ligands for docking were modeled using Avogadro molecular structure editor (v 1.2.0) [44]. The ligands were prepared in Glide “LigPrep” (Schrodinger suite version 2018-4). The structures were generated with possible ionization states at target pH set at 7.0 ± 2.0 using Ionizer, Desalt, and Generate tautomer stereoisomers while retaining the stereo chemical configuration of the input files.

### 4.6. Ligand Docking Simulation

Ligand docking simulations of the modeled structure was performed using Schrödinger Glide, a grid-based ligand docking method (Schrodinger suite version 2018-4) [45,46]. The grid was generated in Glide “Receptor Grid Generation” using the “Centroid of selected residues” set up to enclose the expected binding region based on the predicted active site obtained from discovery studio. No constraints or excluded volumes were specified. The ligands were docked using default glide parameters for scaling of van der Waals radii (vdW) (i.e., scaling factors, 0.8 and partial charge cutoff, 0.15) with precision set at “standard precision” (SP) and no constraints applied. The solutions generated were scored according to Glide scores and Glide energies [45,46,47].

## Figures and Tables

**Figure 1 ijms-22-04493-f001:**
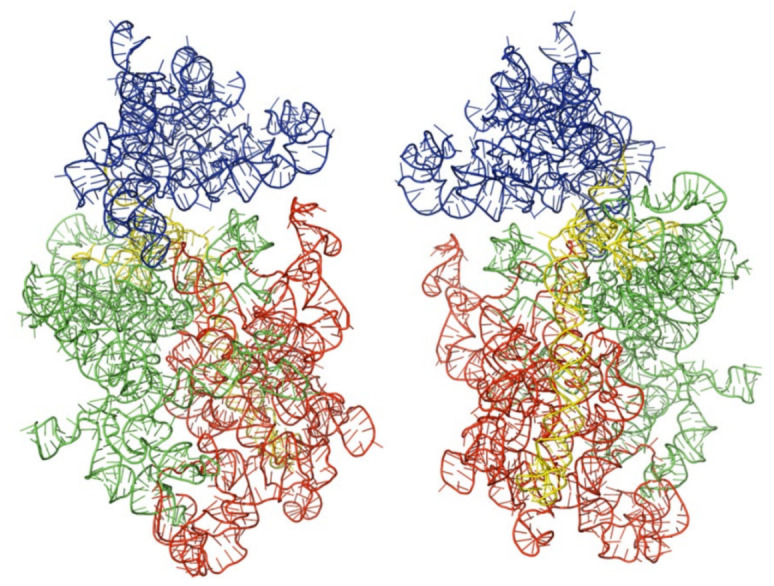
Architectural tertiary structure of *Leishmania major* 18S rRNA front and back view. Shown is the 18S rRNA, colored differently depending with domains (5′major—red, Central—green, 3′major—blue, and 3′minor—yellow).

**Figure 2 ijms-22-04493-f002:**
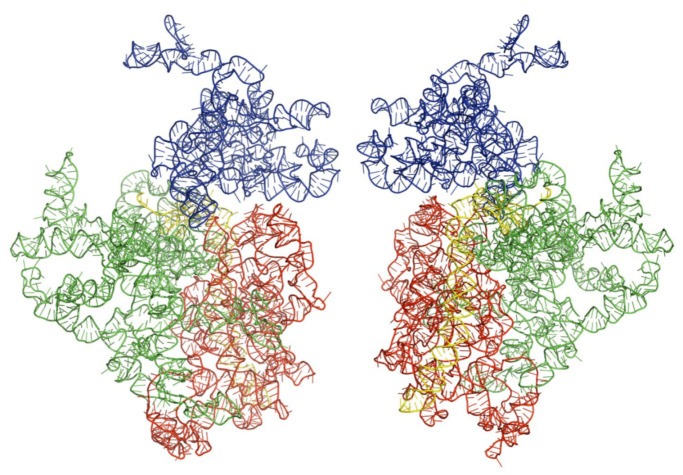
Architectural tertiary structure of *Trypanosoma brucei* 18S rRNA front and back view. Shown is the 18S rRNA, colored differently depending on domains (5′major—red, Central—green, 3′major—blue, and 3′minor—yellow).

**Figure 3 ijms-22-04493-f003:**
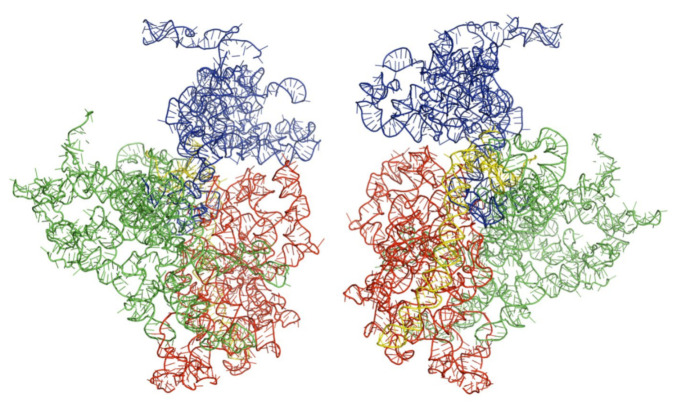
Architectural tertiary structure of *Trypanosoma cruzi* 18S rRNA front and back view. Shown is the 18S rRNA, colored differently depending on domains (5′major—red, Central—green, 3′major—blue, and 3′minor—yellow).

**Figure 4 ijms-22-04493-f004:**
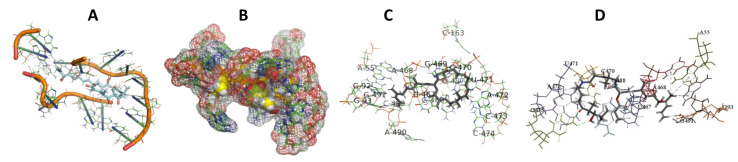
*T. Brucei* bound to Angiolam. (**A**): Best binding pose and nucleotides involved. (**B**): Schematic binding pocket. (**C**): shows the Nucleotides component that re involved in binding. (**D**): Shows the main bonds involved between the compound and the nucleotide component.

**Figure 5 ijms-22-04493-f005:**
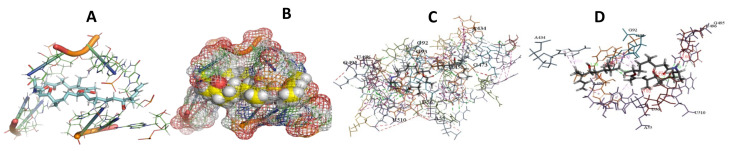
*T. Cruzi* bound to Angiolam. (**A**): Best binding pose and nucleotides involved. (**B**): Schematic binding pocket. (**C**): shows the Nucleotides component that re involved in binding. (**D**): Shows the main bonds involved between the compound and the nucleotide component.

**Figure 6 ijms-22-04493-f006:**
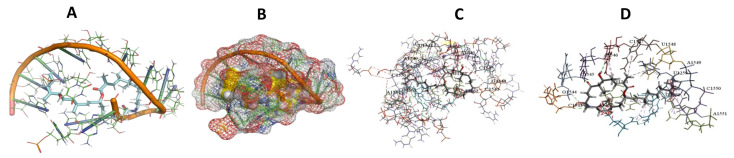
*L. Major* bound to Angiolam. (**A**): Best binding pose and nucleotides involved. (**B**): Schematic binding pocket. (**C**): shows the Nucleotides component that re involved in binding. (**D**): Shows the main bonds involved between the compound and the nucleotide component.

**Figure 7 ijms-22-04493-f007:**
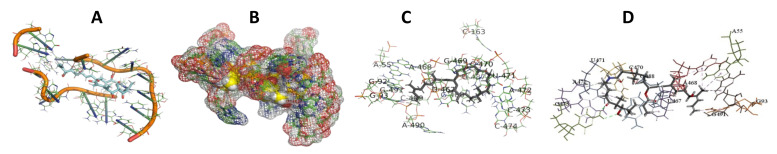
*T. Brucei* + Angiolam. (**A**): Best binding pose and nucleotides involved. (**B**): Schematic binding pocket. (**C**): shows the Nucleotides component that re involved in binding. (**D)**: Shows the main bonds involved between the compound and the nucleotide component.

**Figure 8 ijms-22-04493-f008:**
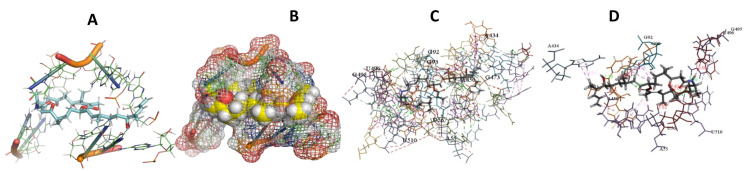
*T. cruzi* + Angiolam (**A**): Best binding pose and nucleotides involved. (**B**): Schematic binding pocket. (**C**): shows the Nucleotides component that re involved in binding. (**D**): Shows the main bonds involved between the compound and the nucleotide component.

**Figure 9 ijms-22-04493-f009:**
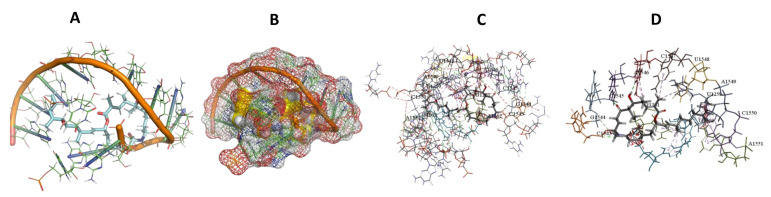
*L. major* + Angiolam. (**A**): Best binding pose and nucleotides involved. (**B**): Schematic binding pocket. (**C**): shows the Nucleotides component that re involved in binding. (**D**): Shows the main bonds involved between the compound and the nucleotide component.

**Figure 10 ijms-22-04493-f010:**
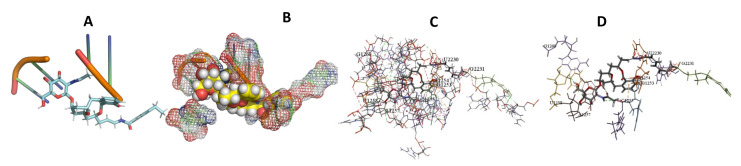
*T. brucei* + Apicuralen (**A**): Best binding pose and nucleotides involved. (**B**): Schematic binding pocket. (**C**): shows the Nucleotides component that re involved in binding. (**D**): Shows the main bonds involved between the compound and the nucleotide component.

**Figure 11 ijms-22-04493-f011:**
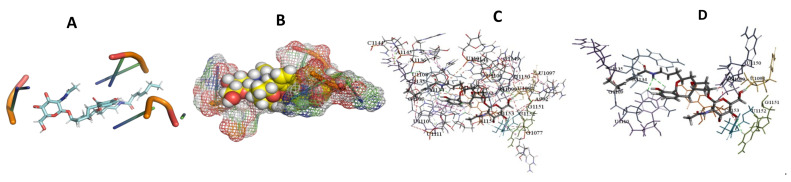
*T. cruzi* + Apicuralen (**A**): Best binding pose and nucleotides involved. (**B**): Schematic binding pocket. (**C**): shows the Nucleotides component that re involved in binding. (**D**): Shows the main bonds involved between the compound and the nucleotide component.

**Figure 12 ijms-22-04493-f012:**
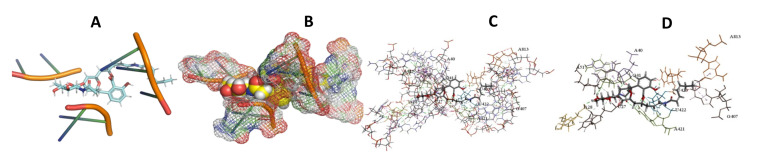
*L. major* + Apicuralen. (**A**): Best binding pose and nucleotides involved. (**B**): Schematic binding pocket. (**C**): shows the Nucleotides component that re involved in binding. (**D**): Shows the main bonds involved between the compound and the nucleotide component.

**Figure 13 ijms-22-04493-f013:**
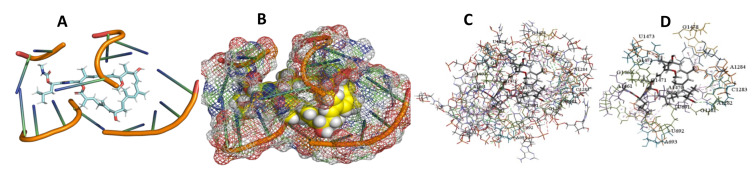
*T. Brucei* + Archazolid. (**A**): Best binding pose and nucleotides involved. (**B**): Schematic binding pocket. (**C**): shows the Nucleotides component that re involved in binding. (**D**): Shows the main bonds involved between the compound and the nucleotide component.

**Figure 14 ijms-22-04493-f014:**
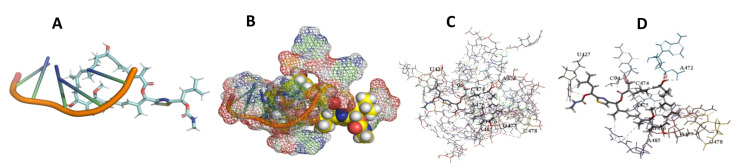
*T. cruzi* + Archazolid. (**A**): Best binding pose and nucleotides involved. (**B**): Schematic binding pocket. (**C**): shows the Nucleotides component that re involved in binding. (**D**): Shows the main bonds involved between the compound and the nucleotide component.

**Figure 15 ijms-22-04493-f015:**
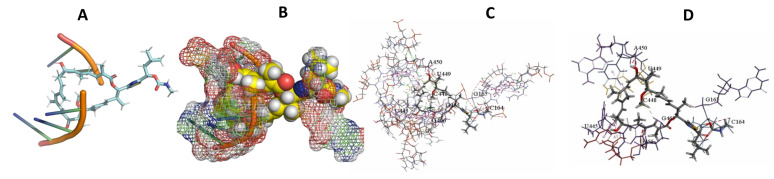
*L. major* + Archazolid. (**A**): Best binding pose and nucleotides involved. (**B**): Schematic binding pocket. (**C**): shows the Nucleotides component that re involved in binding. (**D**): Shows the main bonds involved between the compound and the nucleotide component.

**Figure 16 ijms-22-04493-f016:**
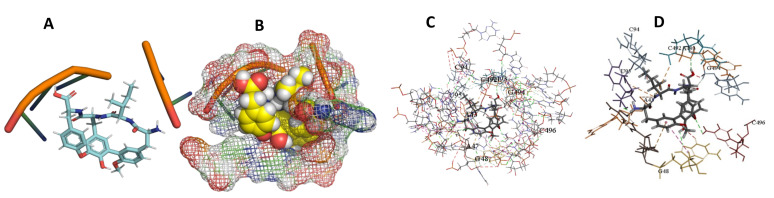
*T. brucei* + Cittilin. (**A**): Best binding pose and nucleotides involved. (**B**): Schematic binding pocket. (**C**): shows the Nucleotides component that re involved in binding. (**D**): Shows the main bonds involved between the compound and the nucleotide component.

**Figure 17 ijms-22-04493-f017:**
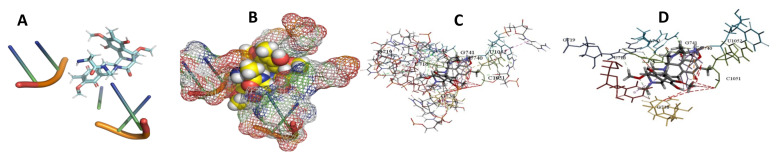
*T. cruzi* + cittilin. (**A**): Best binding pose and nucleotides involved. (**B**): Schematic binding pocket. (**C**): shows the Nucleotides component that re involved in binding. (**D**): Shows the main bonds involved between the compound and the nucleotide component.

**Figure 18 ijms-22-04493-f018:**
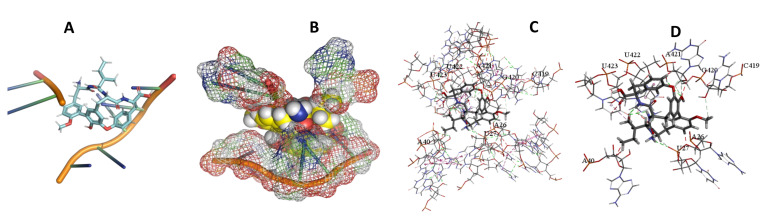
*L. major* + cittilin. (**A**): Best binding pose and nucleotides involved. (**B**): Schematic binding pocket. (**C**): shows the Nucleotides component that re involved in binding. (**D**): Shows the main bonds involved between the compound and the nucleotide component.

**Figure 19 ijms-22-04493-f019:**
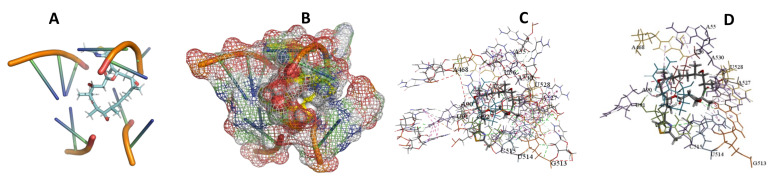
*T. Brucei* + Epothilone. (**A**): Best binding pose and nucleotides involved. (**B**): Schematic binding pocket. (**C**): shows the Nucleotides component that re involved in binding. (**D**): Shows the main bonds involved between the compound and the nucleotide component.

**Figure 20 ijms-22-04493-f020:**
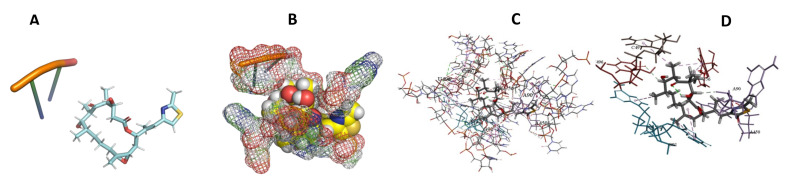
*T. cruzi* + Epothilone. (**A**): Best binding pose and nucleotides involved. (**B**): Schematic binding pocket. (**C**): shows the Nucleotides component that re involved in binding. (**D**): Shows the main bonds involved between the compound and the nucleotide component.

**Figure 21 ijms-22-04493-f021:**
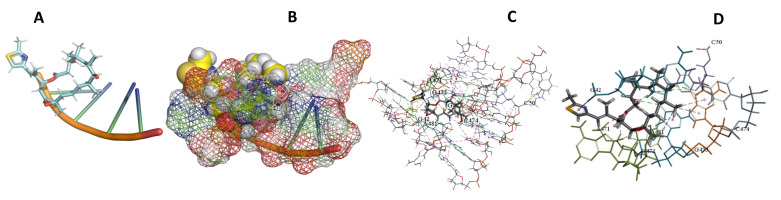
*L. major* + Epothilone. (**A**): Best binding pose and nucleotides involved. (**B**): Schematic binding pocket. (**C**): shows the Nucleotides component that re involved in binding. (**D**): Shows the main bonds involved between the compound and the nucleotide component.

**Figure 22 ijms-22-04493-f022:**
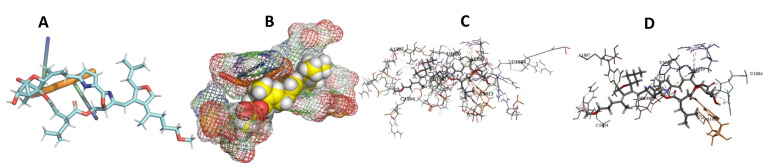
*T. cruzi* + leupyrin. (**A**): Best binding pose and nucleotides involved. (**B**): Schematic binding pocket. (**C**): shows the Nucleotides component that re involved in binding. (**D**): Shows the main bonds involved between the compound and the nucleotide component.

**Figure 23 ijms-22-04493-f023:**
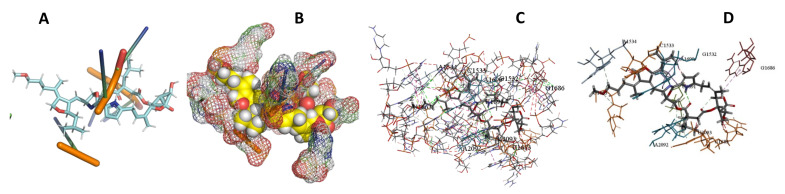
*T. Brucei* + leupyrin. (**A**): Best binding pose and nucleotides involved. (**B**): Schematic binding pocket. (**C**): shows the Nucleotides component that re involved in binding. (**D**): Shows the main bonds involved between the compound and the nucleotide component.

**Figure 24 ijms-22-04493-f024:**
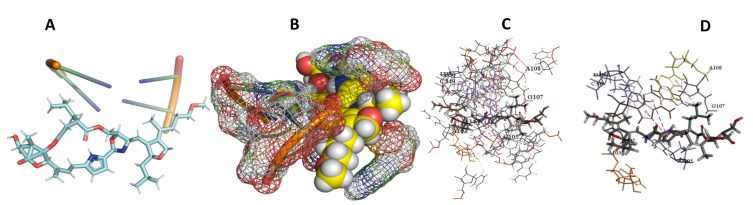
*L. major* + leupyrin. (**A**): Best binding pose and nucleotides involved. (**B**): Schematic binding pocket. (**C**): shows the Nucleotides component that re involved in binding. (**D**): Shows the main bonds involved between the compound and the nucleotide component.

**Figure 25 ijms-22-04493-f025:**
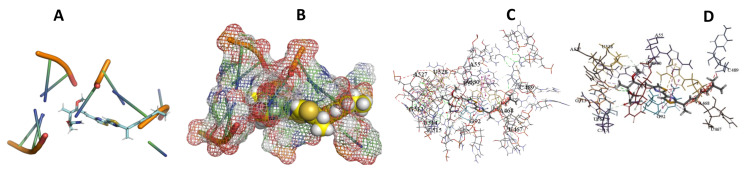
*T. Brucei* + Myxothiazol. (**A**): Best binding pose and nucleotides involved. (**B**): Schematic binding pocket. (**C**): shows the Nucleotides component that re involved in binding. (**D**): Shows the main bonds involved between the compound and the nucleotide component.

**Figure 26 ijms-22-04493-f026:**
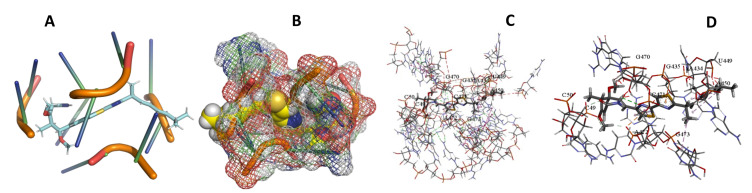
*T. Cruzi* + Myxothiazol. (**A**): Best binding pose and nucleotides involved. (**B**): Schematic binding pocket. (**C**): shows the Nucleotides component that re involved in binding. (**D**): Shows the main bonds involved between the compound and the nucleotide component.

**Figure 27 ijms-22-04493-f027:**
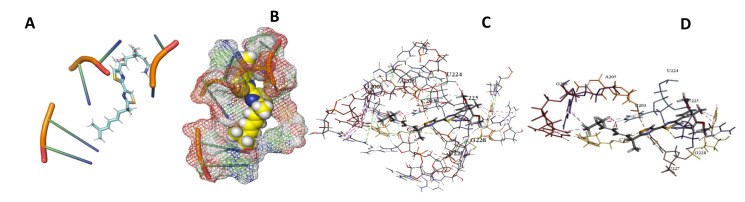
*L. Major* + Myxothiazol. (**A**): Best binding pose and nucleotides involved. (**B**): Schematic binding pocket. (**C**): shows the Nucleotides component that re involved in binding. (**D**): Shows the main bonds involved between the compound and the nucleotide component.

**Figure 28 ijms-22-04493-f028:**
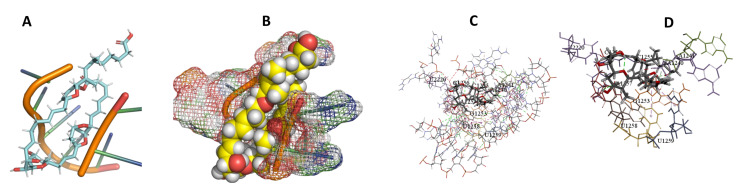
*T. Brucei* + Sorangicin. (**A**): Best binding pose and nucleotides involved. (**B**): Schematic binding pocket. (**C**): shows the Nucleotides component that re involved in binding. (**D**): Shows the main bonds involved between the compound and the nucleotide component.

**Figure 29 ijms-22-04493-f029:**
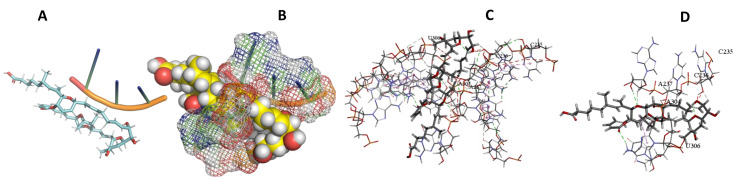
*L. major* + Sorangicin A. (**A**): Best binding pose and nucleotides involved. (**B**): Schematic binding pocket. (**C**): shows the Nucleotides component that re involved in binding. (**D**): Shows the main bonds involved between the compound and the nucleotide component.

**Figure 30 ijms-22-04493-f030:**
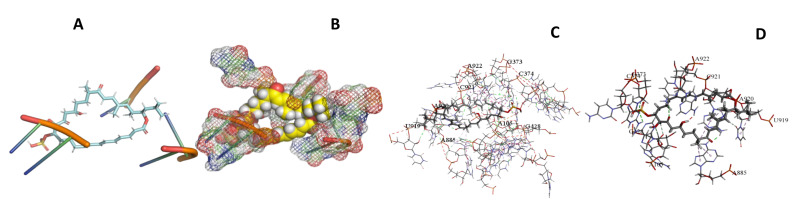
*T. Brucei* + Sulfangolid A. (**A**): Best binding pose and nucleotides involved. (**B**): Schematic binding pocket. (**C**): shows the Nucleotides component that re involved in binding. (**D**): Shows the main bonds involved between the compound and the nucleotide component.

**Figure 31 ijms-22-04493-f031:**
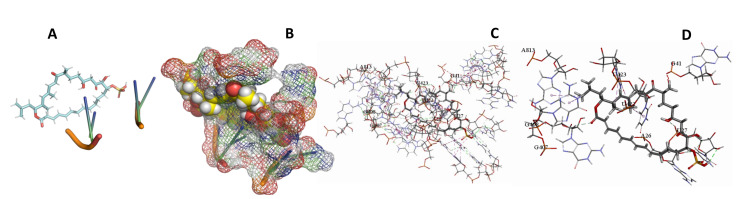
*L. Major* + Sulfangolid A. (**A**): Best binding pose and nucleotides involved. (**B**): Schematic binding pocket. (**C**): shows the Nucleotides component that re involved in binding. (**D**): Shows the main bonds involved between the compound and the nucleotide component.

**Table 1 ijms-22-04493-t001:** The three selected *Trypanosoma brucei*, *Trypanosoma cruzi*, and *Leishmania major* sequences selected for this study. L(3) refers to the cell location, which is the nucleus, RT(4) refers to RNA type R = ribosomal RNA (rRNA), RC refers to the *RNA Class* 16S, Nucleotide size, *Cmp* means *% Complete*, Acc means gene bank accession number, common name and the Phylogenetic Classification, m.

Row #	Organism (2)	L(3)	RT(4)	RC	Size	Cmp	Acc	Common Name	Phylogeny[M] (1)
1	*Trypanosoma brucei*	N	R	16S	2251	100	M12676	kinetoplasts	cellular organisms **…»**
2	*Trypanosoma cruzi*	N	R	16S	2315	100	AF245382	kinetoplasts	cellular organisms **…»**
3	*Leishmania major*	N	R	16S	2203	100	AC005806	kinetoplasts	cellular organisms **…»**

**Table 2 ijms-22-04493-t002:** 18S rRNA Energy Optimization Table obtained from results of RNA 123, which helps minimize the energy from a large positive figure to a more acceptable negative figure that is biologically functional.

Species	Name	18SrRNA.std.egy	18SrRNA.opt.egy
*Leishmania major*	Total Inter energy	908487.3182	−83582.63882
	Total intra energy (-Gamma en	−17864.6686	−17828.1188
	Total Gamma Terms Energy	1746.8654	1743.13568
	Total Gap Geometry Penalty	3108.37922	2746.02559
	Total Restraint Energy	0	3550.87301
	TOTAL STRUCTURE ENERGY	895477.8943	−96921.59635
*Trypanosoma brucei*	Total Inter energy	2021190.532	−102281.5112
	Total intra energy (-Gamma en	71145.86365	10625.04139
	Total Gamma Terms Energy	2357.05629	2277.69707
	Total Gap Geometry Penalty	24166.41849	9500.81691
	Total Restraint Energy	0	8078.41829
	TOTAL STRUCTURE ENERGY	2118859.871	−79877.95586
*Trypanosoma cruzi*	Total Inter energy	7208497.219	−98209.94034
	Total intra energy (-Gamma en	208432.083	−7806.99519
	Total Gamma Terms Energy	2458.82781	2440.13835
	Total Gap Geometry Penalty	35870.86855	12018.7965
	Total Restraint Energy	0	10017.37844
	TOTAL STRUCTURE ENERGY	7455258.998	−91558.00067

**Table 3 ijms-22-04493-t003:** Myxobacteria Compounds with activity on all more negative kinetoplastids, ACE −400.

Compound Name	Compounds with Activity on All More Negative Kinetoplastids, ACE −400		
	*T. Brucei*	*T. Cruzi*	*L. Major*
Angiolam A	−491.7	−673.71	−550.93
Apicularen B	−549.58	−529.41	−585.93
Archazolid A	−516.32	−470.74	−413.53
Cittilin A	−495.42	−529.71	−520.78
Epothilone B	−573.04	−513.65	−346.85
Leupyrin	−598.53	−648.66	−393.82
Myxothiazol	−595.18	−573.36	−449.9
Sorangicin A	−466.93	−466.93	−456.49
Spirangien B	−503.52	−576.45	−516.62
Sulfangolid A	−613.53	−613.53	−643.25

**Table 4 ijms-22-04493-t004:** Showing docking and binding results of the best pose compounds with activity on all more negative kinetoplastids ACE −400.

Compound Name	Compounds with Activity on All More Negative Kinetoplastids ACE −400					
	*T. Brucei*		*T. Cruzi*		*L. Major*	
**Angiolam A**	−491.7	G92,G93,A434,A450,G470,G473,G495,U496,U510	−673.71	A55,U56,G92,G93,A434,A450,G473,G495,U496,U510	−550.93	U1259,G1261,A1262,C1543,G1544,C1545,A1546,C1547,U1548,A1549,C1550,A1551,G1662
Apicularen B	−549.58	G1253,A1254,C1255,A1257,U1258,G1260,U2230,G2231	−529.41	G1109,U1110,A1134,C1135,U1150,G1151,U1152,C1153	−585.93	U27,A28,A40,G41,G407,A421,U422,U423,A813
Archazolid A	−516.32	G690,U691,U692,A693,G1281,A1282,C1283,A1284,G1460,A1461,A1470,G1471,G1472,U1473,G1478	−470.74	C94,U427,A472,C474,A475,G476,G477,C478,A485	−413.53	C164,G165,U445,C448,U449,A450,G465,G466,
Cittilin A	−495.42	A43,A47,G48,C94,U95,C492,A493,G494,C496TTTT	−529.71	U716,G719,G738,U740,G741,A742,C1051,U1052	−520.78	A26,U27,A40,C419,G420,A421,U422,U423,
Epothilone B	−573.04	A55,U56,A90,U91,G92,A468,G513,U514,C515,A527,U528,A530	−513.65	U56,A90,G92,A450,U496,C497,A512	−346.85	G42,C50,A471,G472,G473,C474,A481
Leupyrin	−598.53	G1532,C1533,A1534,U1663,U1683,G1686,A1690,U1691,A2092,U2093	−648.66	C1804,A1807,U1809,A1810,A1813,U1884,U1887	−393.82	A103,C105,G107,A108,A347,C349,U350
Myxothiazol	−595.18	A55,U56,G92,U467,A468,C489,G513,U514,C515,A527,U528,A530	−573.36	C49,C50,A434,G435,U449,A450,G470,C471,A472,G473,	−449.9	C203,A205,G206,C218,U224,C225,U227,G228,
Sorangicin A	−466.93	A1240,A1241,G1253,C1255,C1256,A1257,U1258,U1259,U2220	−466.93		−456.49	C235,C236,A237,A304,U306,
Spirangien B	−503.52	C59,U60,A64,G79,G80,A520,A521,C525,G526	−576.45	A100,G106,G407,G409,C423,G424,A864,A902	−516.62	A1294,C1535,C1536,A1538,A1549,A1642,U1644,U1645,A1689
Sulfangolid A	−613.53	A105,G373,C374,G428,A885,U919,A920,C921,A922	−613.53		−643.25	A26,U27,G41,G407,G408,U422,U423,A813

## Data Availability

Data is contained within the article and supplementary data available if required contact the corresponding author.

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
