# Peer review of "Methods for Identifying Microbial Natural Product Compounds that Target Kinetoplastid RNA Structural Motifs by Homology and De Novo Modeled 18S rRNA"

_ijms, 2021, doi:10.3390/ijms22094493_

Round 1
Reviewer 1 Report
In the present work, the authors investigate a methods for identifying microbial natural product compounds that target kinetoplastid RNA structural motifs by homology and de novo modeled 18S rRNA. This research is interesting.
TITLE
The paper title is well stated, it is informative and concise.
ABSTRACT AND INTRODUCTION
Abstract and Inrtoduction were well written.
MATERIAL AND METHODS
Material and research methods are presented appropriately and clearly. Experimental setup and the description in the methods section are well structured, precise enough, clearly described.
RESULTS
The results obtained in this study are interesting. The results are presented appropriately and clearly. In my opinion, all Figs should be included as a supplementary data.
DISCUSSION
In general, the discussion of results is correct.
LITERATURE
The items of literature included in the paper are adequate to the subject of the paper.
Author Response
The Author: We further refined the Major and the minor Correction that had been suggested by the other reviewer to improve on the article submitted
Reviewer 2 Report
In this paper, the authors reported the in silico method combined with both computational biology and structure prediction tools for drug discovery of parasites.
This is an interesting topic for development of pan-active treatments against kinetoplastids, and the manuscript is well analyzed. However, I wish the authors address the following comments. It should be necessary to show additional experiments.
Major comment:
The authors must experimentally determine the actual binding constants in the target RNA and small molecule combinations (only one combination is enough) to evaluate the certainty and usefulness of the in silico method reported here.
Minor comments:
Line 257, font (color and bold type) caption in fig.1 should be corrected.
Line 260, “Architectural tertiary structure of Trypanosoma brucei 18S r” in fig.2 should be corrected to normal font type (not bold).
In table 2, “Trypanosoma” and “895477.8943” etc. are indicated as bold type. Why are those letters written in bold? If it's important, add a description to the caption.
Table captions should be written ABOVE the table.
Line 308, “figure 5-32” should be corrected to “figure 4-31”.
Line 313, Delete “Natural compounds binding sites on Various Kinetoplastids”. If you need it, add table caption or main text.
Add description of A, B, C and D in fig. 4-31 (only one description is enough).
Line 317, Delete “at” in fig.5 caption.
Line 325, “figure 3” should be corrected to “figure 9”.
References should be described as follows,
******Journal Articles:
1. Author 1, A.B.; Author 2, C.D. Title of the article. Abbreviated Journal Name Year, Volume, page range.
******Books and Book Chapters:
2. Author 1, A.; Author 2, B. Book Title, 3rd ed.; Publisher: Publisher Location, Country, Year; pp. 154–196.
3. Author 1, A.; Author 2, B. Title of the chapter. In Book Title, 2nd ed.; Editor 1, A., Editor 2, B., Eds.; Publisher: Publisher Location, Country, Year; Volume 3, pp. 154–196.
Author Response
Major comment:
The authors further described and cited a previous experimental validation technique that had been used for the entire ribosome to determine the actual binding constants in the target RNA and small molecule combinations (). In addition, explained and evaluated the certainty and usefulness of the in silico method in drug discover and reported the quick process of screening known rRNA structures against an array or database of natural compound. In the recommendation further work has been suggested where synthesis of the specific aptamer could be reacted to the natural compounds to generate chromatograms to further analyse the natural compound to be assayed as probable therapeutics
Minor comments:
Line 257, font (color and bold type) caption in fig.1 was corrected.
Line 260, Corrected as suggested by the reviewer
In table 2, The formatting changes were done and more description of the table was explained
Table captions were written ABOVE the table.
Line 308, “figure 5-32” was corrected to “figure 4-31”.
Line 313, “Natural compounds binding sites on Various Kinetoplastids” was retained and main text was added to describe A, B, C and D
Line 317, “at” was deleted in fig.5 caption.
Line 325, “figure 3” was corrected to “figure 9”.
References were corrected as the reviewer has suggested described as follows
Round 2
Reviewer 2 Report
I have checked the revised manuscript. I am satisfied with the response of the authors and the revisions. Therefore, I recommend acceptance of this manuscript.
Minor comments:
Again, carefully check the format of all of references.
For example,
1. Stuart, Ken, Reto Brun, Simon Croft, Alan Fairlamb, Ricardo E. Gürtler, Jim McKerrow, Steve Reed, and Rick Tarleton. "Kinetoplastids: related protozoan pathogens, different diseases." The Journal of clinical investigation 118, no. 4 (2008): 1301-1310.
should be corrected to
1. Stuart, K.; Brun, R.; Croft, S.; Fairlamb, A.; Gürtler, R.E.; McKerrow, J.; Reed, S.; Tarleton, R. Kinetoplastids: related protozoan pathogens, different diseases. J. Clin. Invest. 2008, 118, 1301-1310.
and,
11. Angelbello, Alicia J., Suzanne G. Rzuczek, Kendra K. Mckee, Jonathan L. Chen, Hailey Olafson, Michael D. Cameron, Walter N. Moss, Eric T. Wang, and Matthew D. Disney. "Precise small-molecule cleavage of an r(CUG) repeat expansion in a myotonic dystrophy mouse model."Proceedings of the National Academy of Sciences 116, no. 16 (2019): 7799-7804.
should be corrected to
11. Angelbello, A.J.; Rzuczek, S.G.; Mckee, K.K.; Chen, J.L.; Olafson, H.; Cameron, M.D.; Moss, W.N.; Wang, E.T; Disney, M.D. Precise small-molecule cleavage of an r(CUG) repeat expansion in a myotonic dystrophy mouse model. Proc. Natl. Acad. Sci. U. S. A. 2019, 116, 7799-7804.
and,
13. Disney, M.D., Targeting RNA with small molecules to capture opportunities at the intersection of chemistry, biology, and medicine. Journal of the American Chemical Society, 2019. 141(17): p. 6776-6790.
should be corrected to
13. Disney, M.D. Targeting RNA with small molecules to capture opportunities at the intersection of chemistry, biology, and medicine. J. Am. Chem. Soc. 2019, 141, 6776-6790.
etc.
Search Tool for journal abbreviation is follows.
https://cassi.cas.org/search.jsp
If you can not find journal abbreviation, also see PubMed (https://pubmed.ncbi.nlm.nih.gov/).